# Low-Molecular-Weight Heparins (LMWH) and Synthetic Factor X Inhibitors Can Impair the Osseointegration Process of a Titanium Implant in an Interventional Animal Study

**DOI:** 10.3390/medicina58111590

**Published:** 2022-11-03

**Authors:** Dragos Apostu, Bianca Berechet, Daniel Oltean-Dan, Alexandru Mester, Bobe Petrushev, Catalin Popa, Madalina Luciana Gherman, Adrian Bogdan Tigu, Ciprian Ionut Tomuleasa, Lucian Barbu-Tudoran, Horea Rares Ciprian Benea, Doina Piciu

**Affiliations:** 1Department of Orthopedics, Traumatology and Pediatric Orthopaedics, University of Medicine and Pharmacy Cluj-Napoca, 400347 Cluj-Napoca, Romania; 2Department of Gastroenterology, “Octavian Fodor” Institute of Gastroenterology and Hepatology, 400347 Cluj-Napoca, Romania; 3Department of Oral Health, University of Medicine and Pharmacy Cluj-Napoca, 400012 Cluj-Napoca, Romania; 4Department of Materials Science and Engineering, Technical University of Cluj-Napoca, 400114 Cluj-Napoca, Romania; 5Experimental Center, University of Medicine and Pharmacy Cluj-Napoca, 400012 Cluj-Napoca, Romania; 6Research Center for Advanced Medicine—MedFuture, Department of Translational Medicine, University of Medicine and Pharmacy Cluj-Napoca, 400347 Cluj-Napoca, Romania; 7Department of Hematology, University of Medicine and Pharmacy Cluj-Napoca, 400012 Cluj-Napoca, Romania; 8Department of Hematology, Ion Chiricuta Clinical Cancer Center, 400015 Cluj-Napoca, Romania; 9Medfuture Research Center for Advanced Medicine, University of Medicine and Pharmacy Cluj-Napoca, 400012 Cluj-Napoca, Romania; 10Electron Microscopy Center, Faculty of Biology and Geology, Babes-Bolyai University, 400006 Cluj-Napoca, Romania; 11Nuclear Medicine Department, University of Medicine and Pharmacy Cluj-Napoca, 400012 Cluj-Napoca, Romania

**Keywords:** aseptic loosening, cementless hip arthroplasty, enoxaparin, fondaparinux, implant osseointegration, nadroparin

## Abstract

*Background and objectives:* Cementless total hip arthroplasty is a common surgical procedure and perioperative thromboprophylaxis is used to prevent deep vein thrombosis or pulmonary embolism. Osseointegration is important for long-term implant survival, and there is no research on the effect of different thromboprophylaxis agents on the process of osseointegration. *Materials and Methods:* Seventy rats were allocated as follows: Group I (control group), Group II (enoxaparin), Group III (nadroparin), and Group IV (fondaparinux). Ovariectomy was performed on all subjects, followed by the introduction of an intramedullary titanium implant into the femur. Thromboprophylaxis was administered accordingly to each treatment group for 35 days postoperatively. *Results:* Group I had statistically significantly lower anti-Xa levels compared to treatment groups. Micro-CT analysis showed that nadroparin had lower values compared to control in bone volume (0.12 vs. 0.21, *p* = 0.01) and percent bone volume (1.46 vs. 1.93, *p* = 0.047). The pull-out test showed statistically significant differences between the control group (8.81 N) compared to enoxaparin, nadroparin, and fondaparinux groups (4.53 N, 4 N and 4.07 N, respectively). Nadroparin had a lower histological cortical bone tissue and a higher width of fibrous tissue (27.49 μm and 86.9 μm) at the peri-implant area, compared to control (43.2 μm and 39.2 μm), enoxaparin (39.6 μm and 24 μm), and fondaparinux (36.2 μm and 32.7 μm). *Conclusions:* Short-term administration of enoxaparin, nadroparin, and fondaparinux can reduce the osseointegration of titanium implants, with nadroparin having the most negative effect. These results show that enoxaparin and fondaparinux are preferred to be administered due to a lesser negative impact on the initial implant fixation.

## 1. Introduction

Total hip replacement is a common procedure performed worldwide to treat hip osteoarthritis. This pathology affects 10 to 13% of people over 60 years old [1]. Total hip replacement implants are made from titanium alloys. The most frequently used titanium alloy in total hip replacements is Ti_90_Al_6_V_4_, consisting of titanium, aluminium and vanadium. Although offering good results overall, complications of total hip replacement exist. The most frequent late complication of this type of this surgical procedure is a deficient implant fixation, called aseptic loosening, which leads to increased pain and disability [2]. Patients affected by this complication cannot weight-bear on the affected limb, thus leading to an important functional deficit. When aseptic loosening is present, revision surgery is required, which is expensive for the healthcare system [3]. Moreover, it is technically demanding, requires an experienced team, and is usually performed in tertiary-care hospitals. Additionally, the revision of the total hip replacement often results in a lower patient satisfaction rate than the primary hip replacement [4].

Aseptic loosening can be prevented with a more enhanced osseointegration process, represented by bone apposition at the titanium surface of the total hip arthroplasty implant [5,6]. This complex process is dependent on the processes of bone formation, performed by osteoblasts, and bone resorption, performed by osteoclasts. The process of osseointegration is regulated by many cellular pathways which modulate the activity of osteoblasts and osteoclasts [6]. Osteoblasts arise from mesenchymal stem cells (MSC) under the influence of cytokines, such as tumour necrosis factor (TNF) alpha, interleukin (IL) 1, IL-6 and IL-11. On the other hand, osteoclastogenesis is positively modulated by macrophage colony-stimulating factor (M-CSF) and transforming growth factor beta (TGF-β1). The more active the osteoblasts are compared to osteoclasts, the stronger the implant fixation will be and the lower the risk of aseptic loosening. The process of osseointegration is similar in the case of titanium intramedullary implants and titanium dental implants [7,8,9]. Our study group has proven that osseointegration can be influenced by many factors, including systemic drugs [10].

Low-molecular-weight heparins (LMWH) and synthetic factor X inhibitors are routinely administered postoperatively to prevent deep vein thrombosis (DVT) and pulmonary embolism. Among the most commonly used drugs to prevent DVT postoperatively following total hip replacement are enoxaparin, nadroparin, and fondaparinux. 

Previous studies by Kock et al. and Osip et al. showed that LMWHs could inhibit osteoblastogenesis during in vitro experimental studies [11,12,13,14]. These studies were the first to prove that low-molecular-weight heparins have an impact on bone metabolism. One explanation is that LMWHs can alter the function of the cytokines involved in osteoblastogenesis and osteoclastogenesis [15]. As a result, the whole osseointegration process can be affected [15]. Numerous in vivo studies were performed to study the effects of LMWHs on bone biology. Enoxaparin, dalteparin, nadroparin and tinzaparin were shown to increase osteoclastogenesis and bone resorption by modulating M-CSF and TGF- β1, [16,17,18,19,20,21,22]. Additionally, researchers studied the effects of LMWHs on bone metabolism in the case of fracture healing. A study performed by Strett et al. showed that enoxaparin attenuated the bone repair process compared to the control group [23]. The result is confirmed by a more recent study by Li et al., which concluded that enoxaparin suppresses osteoblastogenesis [24]. On the other hand, other studies showed that LMWHs did not impair the fracture process [25,26]. 

Although previous studies showed that LMWHs have an impact on bone biology, the overall effect is still controversial. Moreover, we did not find any study to test the effects of LMWHs on the process of osseointegration of the titanium implant. We consider that in vivo studies are essential to test the osseointegration process of titanium implants because there is an implication of osteoclasts, osteoblasts and the titanium surface, which are impossible to replicate in the case of in vitro studies. We also consider it essential to know whether thromboprophylaxis agents can impair the titanium implant osseointegration. Clinical trials are challenging to be performed due to a lack of specific examinations for the osseointegration process in the clinical setting, which are available in an animal model. Moreover, the long follow-up to study the rate of aseptic loosening makes clinical trials more difficult to perform.

This study aims to test, for the first time in the literature, the impact of thromboprophylaxis agents on the process of osseointegration in vivo. For this study, we tested three of the most commonly used drugs in DVT prophylaxis: enoxaparin, nadroparin, and fondaparinux, in terms of early implant fixation in vivo. 

## 2. Materials and Methods

### 2.1. Animal Model

The study received approval from the Ethical Commission of the local university (no. 210/02/04/2020). The experiments were performed at the Center of Experimental Medicine Cluj-Napoca and according to the European guidelines (directive 2010/63/EU). A total of 70 female albino Wistar rats of 8–10 weeks old and with a weight of 190 ± 30 mg were used. The animals were raised at the same animal facility without any genetic modification, while food and water were provided ad libitum. A veterinary doctor checked all of the subjects to be enrolled in the study to be clinically healthy. The subjects were randomized into four groups: Group I (OVX group, *n* = 22), Group II (OVX + enoxaparin, *n* = 16), Group III (OVX + nadroparin, *n* = 16), and Group IV (OVX + fondaparinux, *n* = 16). 

### 2.2. Ovariectomy (OVX) Procedure

The procedures were performed on all subjects at the time of enrolment, and each subject’s weight was determined preoperatively. General anaesthesia was induced with a mixture of 80–100 mg/kg of ketamine and 10–12.5 mg/kg of xylazine injected intraperitoneally. Skin preparation using betadine and sterile draping was performed. Following an abdominal midline incision, the ovaries were identified and excised with electrocautery (Figure 1a and Figure 2). The abdominal wall was sutured, and a local antibiotic was applied. Postoperatively, analgesics were provided in the drinking water.

### 2.3. Intramedullary Nail

Three months after the ovariectomy procedure, bilateral femoral intramedullary nailing was performed in all subjects under general anaesthesia. Both legs were prepared, and sterile draping was performed. The femoral condyles were palpated, and using a sterile 18-gauge needle, and the femoral canal was opened percutaneously at the level of the femoral notch (Figure 1b and Figure 2). Then, Ti_90_Al_6_V_4_ alloy nails (Goodfellow Cambridge Ltd., Huntingdon, UK) with a diameter of 1 mm and 20 mm in length were introduced into the femur (Figure 1c). There was no need for a suture since the technique was entirely percutaneous. A local antibiotic was applied to the insertion site. Postoperatively, analgesics were provided in the drinking water.

### 2.4. Treatment

Starting on day one postoperatively, the subjects were weighted daily, and treatment was administered subcutaneously according to each group (Figure 2). Group I received saline subcutaneously daily for 35 days. Group II received enoxaparin 1 mg/kg subcutaneously daily for 35 days. Group III received receive nadroparin 10 mg/kg subcutaneously daily for 35 days and group IV received receive fondaparinux 0.1 mg/kg subcutaneously daily for 35 days. 

### 2.5. Collection of Samples

Three months after the intramedullary nailing, blood samples were harvested. Under general anaesthesia, the subjects were euthanised. We looked for uterus atrophy and the absence of ovaries. Moreover, the bilateral femoral bones were collected and placed in 10% formaldehyde. The femurs underwent histological, micro-CT, and mechanical pull-out test examinations.

### 2.6. Serum Analysis

Using the ELISA method, we tested the rat coagulation factor Xa (NovusBiologicals^®^, Cambridge, UK). The calibration was performed using the following concentrations: 40 ng/mL, 20 ng/mL, 10 ng/mL, 5 ng/mL, 2.5 ng/mL, 1.25 ng/mL and 0.63 ng/mL. The samples were diluted five times. We used the TECAN Spark 10M (TECAN, Grödig, Austria) microplate reader.

### 2.7. Micro-CT Examination

Bruker Skyscan 1172^®^ (Billerica, MA, USA) with a 50 mm image field width and 11 Mp X-ray camera was used at a resolution of 2000 × 2000 px for the micro-CT scanning. We analysed a region of interest of a round shape, and a diameter of 120 mm centred on the implant. The length of the area of interest was 600 slices (8.1 mm) starting from the distal metaphysis proximally. The Bruker CTAn^®^ v.1.18. software was used to calculate bone volume (BV), percent bone volume (BV%), bone surface (BS), bone surface/volume ratio (BS/VR), tissue surface (TS), mean total cross-sectional bone area, trabecular number (TN), cross-sectional thickness, and trabecular diameter (TD). The measurements were performed while the examinator was unaware of group allocation. 

### 2.8. Mechanical Pull-Out Test

An osteotomy of the femoral diaphysis in the proximal one third was sequentially performed until 5 mm of the intramedullary nail was exposed. The nail was tightened in a pneumatic grip, and the Zwick/Roell Z005^®^ (Ulm, Germany) tensile testing device with a maximum test load of 5 kN was used to measure the force needed for nail extraction. The forces were measured in newtons at a low speed of 1 mm/min. The measurements were performed while the examinator was unaware of group allocation.

### 2.9. Histological Examination

Following the mechanical pull-out test, the femoral bones were decalcified and sectioned longitudinally along the implant site. Hematoxylin–eosin and Tricom Masson stainings were performed and analysed with a Leica DM750^®^ (Wetzlar, Germany) microscope. We used ImageJ^®^ software for the morphometric measurements. The thickness of the cortical bone at the peri-implant site was measured at five different points at about 20-0 µm apart. The same method was applied to measure the fibrous tissue at the peri-implant site. Two independent measurements were performed by different examinators who did not know the allocation within the groups, and the average was noted.

### 2.10. SEM/EDX Analysis

We tested a total of 12 samples equally divided into groups. The samples were fixed in glutaraldehyde (2.7% in PBS), dehydrated in alcohol, and infused with hexamethyldisilane. After sputter-coating with 7 nm of gold in an Agar Automatic Sputter-Coater B7341 (Essex, UK), samples were examined in a Hitachi SU8230 HRCFEG SEM (Tokyo, Japan).

### 2.11. Statistical Analysis

The sample size was calculated during the study design using the StatMate^®^ software. For the sample size calculation, we used the results obtained from previous studies on titanium implant osseointegration following systemic administration [27,28]. The type I/II error rates we used during calculations were alpha values of 0.05 and power of the study of 80%. Moreover, we assumed a 20% mortality rate due to the two surgical interventions and general anaesthesia. The statistical analysis was performed using the GraphPad Prism 6.0^®^ software, and we calculated means, standard deviations, frequencies, percentages, and correlation tests. The distribution was calculated using the Shapiro–Wilk test. The results were considered statistically significant if the *p*-value was less than 0.05. 

## 3. Results

Of the seventy subjects included in the study, nine died throughout the process: seven died due to anaesthesia, and two died from infection. Five of the subjects belonged to Group I, one subject belonged to Group II, and three subjects belonged to Group IV. 

### 3.1. Weight

All of the subjects were weighed during the study. The average weight before the ovariectomy procedure was 216 ± 22 g. The mean weight according to each group are as follows: 209.4 ± 17.5 g (Group I), 221.5 ± 23 g (Group II), 216 ± 25 g (Group III), and 222 ± 21 g (Group IV). Group I had a statistically significant lower weight compared to Groups II, III and IV (*p* < 0.05).

At the end of the study, the mean weight according to each group were as follows: 215 ± 16.5 g (Group I), 230 ± 20 g (Group II), 225 ± 27 g (Group III), and 229 ± 21 g (Group IV). Additionally, Group I had a statistically significantly lower weight than Groups II, III and IV (*p* < 0.05). Nevertheless, all of the subjects increased in weight during the study.

### 3.2. Serum Analysis

The rat coagulation factor Xa was calculated in all of the subjects at the end of the study (n = 61), as follows: 17 subjects in Group I, 15 subjects in Group II, 16 subjects in Group III and 13 subjects in Group IV. The results are available in Figure 3. Group I had statistically significantly lower levels than Groups II, III and IV (*p* < 0.001). There were no statistically significant differences between Groups II, III and IV. 

### 3.3. Micro-CT Examination

A total of 12 samples were analysed, equally divided within the groups. The results of the calculate bone volume (BV), percent bone volume (BV%), bone surface (BS), bone surface/volume ration (BS/VR), tissue surface (TS), mean total cross-sectional bone area, trabecular number (TN), cross-sectional thickness and trabecular diameter (TD) are available in Table 1. The only statistically significant differences observed were between Group I and Group III in terms of bone volume (*p* = 0.001) and percent bone volume (*p* = 0.047). Images obtained during micro-CT examinations are available in Figure 4.

### 3.4. Mechanical Pull-Out Test

The mechanical pull-out test was performed in all of the subjects (*n* = 61). The average values of the maximum force needed to extract the intramedullary nail are available in Figure 5. The control group had the highest average force needed for intramedullary nail extraction, followed by enoxaparin. There were statistically significant differences between Group I and Group III (*p* = 0.01), Group I and Group IV (*p* = 0.03), as well as between Group I and Group II (*p* = 0.04). There were no statistically significant differences between the treatment groups.

### 3.5. Histological Analysis

Histological analysis was performed on all of the subjects within the study (*n* = 60), except for one subject in Group IV whom we excluded after SEM/EDX analysis due to higher concentrations of calcium and phosphorus on the implant after its removal. Images from the measurements of the fibrous tissue and bone tissue are seen in Figure 6 and Figure 7. 

In terms of cortical bone surrounding the implant site, the nadroparin group had the lowest width (see Figure 8). The only statistically significant result was obtained between Group I and Group III (*p* = 0.0002). 

Regarding fibrous tissue surrounding the implant, the nadroparin group had the highest values (see Figure 9). The only statistically significant results were between Group II and Group III (*p* = 0.02) and between Group III and Group IV (*p* = 0.04).

### 3.6. SEM/EDX Analysis

The SEM/EDX analysis was performed on a number of three titanium implants within each group, resulting in a total of twelve implants. Except for one case, the maximum Wt% of both calcium and phosphorus was 1.3%. There were no statistically significant differences between groups in terms of calcium and phosphorus concentrations. 

Moreover, the EDX analysis of the titanium, aluminium, and vanadium showed an average concentration of 88.6%, 7.3%, and 3.8%, respectively. The images obtained from the SEM/EDX analysis are available in Figure 10.

There was one case belonging to Group IV (Figure 11), where the Wt% of calcium was 52.9% and 25.4% in the case of phosphorus. 

## 4. Discussion

To our knowledge, this is the first in vivo study to test and compare the effect of enoxaparin, nadroparin, and fondaparinux on the process of titanium implant osseointegration. 

The rat animal model was used due to the high resemblance of bone metabolism compared to humans. They also represent the most commonly used animal model for the study of osseointegration. In clinical practice, patients requiring total hip arthroplasty are frequently osteoporotic and have a deficient bone metabolism [29]. Therefore, we decided to perform an ovariectomy to induce osteoporosis in our animal model. Additionally, when an osteoporotic animal model is used, there is a higher impact on the treatment involved, meaning a lower number of subjects is needed to obtain statistically significant results [30]. In the literature, osteoporosis is generally accepted to be obtained at three months; therefore, we performed the intramedullary implantation of the nails at 12 weeks. For this study, we used Ti_90_Al_6_V_4_ alloy nails which resemble the alloy of cementless total hip arthroplasties. Moreover, throughout the study, we had no deaths due to pulmonary embolism because the surgical intervention was minimally invasive and the subjects could completely weight bear after the surgery. 

The treatment was administered for 35 days after titanium nail implantation, which is the current protocol in our country. The doses were obtained from different studies in which deep vein thrombosis prophylaxis was performed in rat animal models using the three types of treatment [31,32,33]. In order to test the efficiency of the treatment administration, rat coagulation factor Xa was determined. Additionally, we performed the SEM/EDX analysis of the extracted implants to identify any bone tissue extracted along with the implant.

The anti-Xa levels were statistically significantly increased in the treatment groups compared to the control group, showing a good treatment administration. Nevertheless, we do not have an established therapeutic range in the animal model. As a result, the serum analysis does not provide information about the dosing. 

The micro-CT analysis showed that nadroparin statistically significantly has a lower bone volume around the implant compared to the control group. Moreover, there were no statistically significant differences between the treatment groups. These results show the effect of these drugs in the osseointegration process of titanium implants. Moreover, despite other differences observed, the sample size in the case of micro-CT examinations was insufficient to provide other statistically significant results.

The pull-out test is an essential tool to assess the strength of the osteointegration process because it tests both the quantity and the quality of the implant fixation. During the pull-out test, all treatment groups showed a significantly lower force needed for implant extraction compared to the control group. This result shows a stronger implant fixation in the absence of treatment. 

The histological analysis found a significantly lower cortical bone mass around the implant site in the case of nadroparin compared to the control group. Despite enoxaparin and fondaparinux also showing an overall lower cortical bone mass compared to the control, as well, the results were not statistically significant. Nevertheless, the biggest difference was when comparing the fibrous tissue around the implant site, which was significantly higher in the nadroparin group than in all other groups. The histological results show that nadroparin significantly reduces implant fixation than in all other groups. 

We also performed the SEM/EDX analysis to determine whether bone tissue was still present on the implant surface after its extraction from the femoral bone during the pull-out test. The EDX analysis only showed small concentrations of calcium and phosphorus in all of the implants, except in one case, where there was a high concentration of calcium and phosphorus and was later excluded from the histological examination. These results show that most implants did not contain relevant concentrations of calcium or phosphorus following implant removal.

We have found no similar studies to compare our study’s results. Nevertheless, other studies on bone metabolism showed that enoxaparin and nadroparin increase osteoclastogenesis, thus leading to an increased bone resorption process [16,17,18,19,20,21,22]. These results are according to our study, where the osseointegration process is impaired by enoxaparin and nadroparin. Other studies on fracture healing have shown that enoxaparin can impair bone growth compared to control, a result related to our study due to a deficient bone metabolism [23].

Bruker Skyscan 1172 micro-CT is a microfocus X-ray microtomography optimized for small samples offering good precision for osseointegration examination. The error and tolerance are insignificant since a standard method of determining the region of interest was used in all samples. The Zwick/Roell Z005^®^ testing machine has a machine compliance correction, offering real-time modifications for the highest possible level of precision.

The study also has some limitations. The main limitation of our study is the inability to perform the histological analysis with the implant in situ. This could provide a better assessment of the histological bone–implant contact. During the implant removal, even though it is performed at low speeds, a quantity of bone and fibrous tissue could still be attached to the implant and therefore provide deficient information when the bone specimens are analysed. In order to test this hypothesis, we performed an SEM/EDX analysis which showed that only a small quantity of calcium and phosphorus had been removed along with the implants, with only one exception, which was later excluded. This result provides sufficient information to state that the implant removal during the pull-out test at low speed does not affect the histological analysis of the implant site. 

Another limitation is the lack of a known therapeutic range for the anti-Xa coagulation factor level for us to know whether or not the dosing was correct. This limitation could influence the study’s results due to potentially different concentrations of drugs. Additionally, a limitation of our study is the relatively low number of subjects for micro-CT analysis and SEM/EDX analysis.

Our study has shown, for the first time in the literature, that some low-molecular-weight heparins (LMWH) and synthetic factor X inhibitors can affect the process of osseointegration in vivo. An impaired osseointegration process can lead to aseptic loosening of cementless total hip arthroplasties. Our study’s results are significant, especially due to the long-term follow-up needed to determine the association between aseptic loosening and the type of thromboprophylaxis drug in a clinical setting. We found no clinical studies to determine this association and we recommend further studies be performed in this direction, since our study has proven the implication of thromboprophylaxis agents on the process of osseointegration.

## 5. Conclusions

Short-term administration of enoxaparin, nadroparin, and fondaparinux can reduce the osseointegration of titanium implants, while nadroparin resulted in the highest quantity of fibrous tissue and the lowest quantity of cortical bone tissue surrounding the implant site. Further clinical research is needed to test the influence of thromboprophylaxis agents on the process of osseointegration.

## Figures and Tables

**Figure 1 medicina-58-01590-f001:**
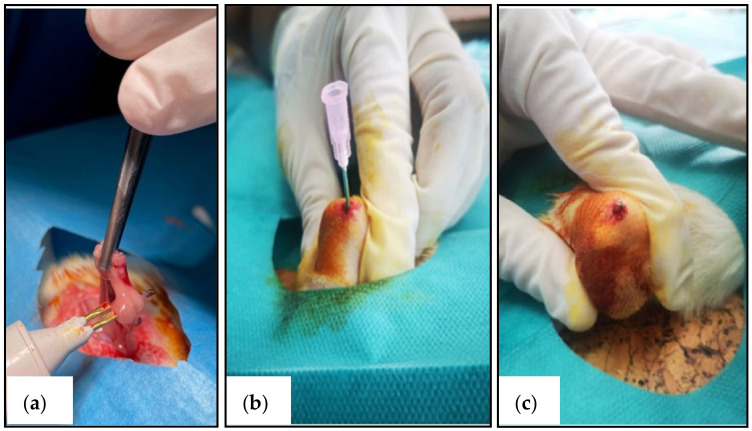
Intraoperative images: (**a**) Ovariectomy procedure; (**b**) Opening of the femoral canal; (**c**) Implantation of the titanium intramedullary nail.

**Figure 2 medicina-58-01590-f002:**
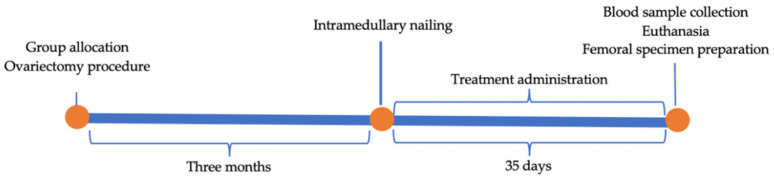
Project’s timeline: group allocation, ovariectomy procedures, intramedullary nailing, treatment administration, blood sample collection, euthanasia, and femoral specimen preparation.

**Figure 3 medicina-58-01590-f003:**
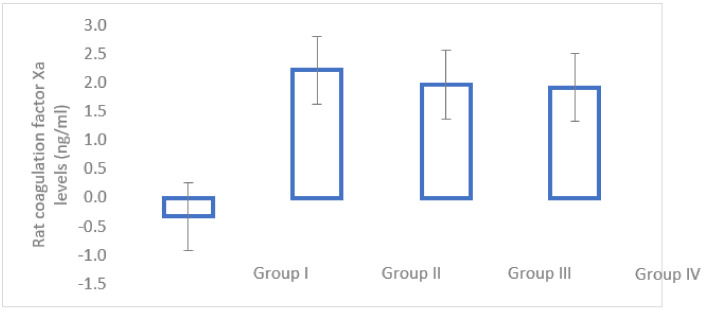
Rat coagulation factor Xa levels at the end of the study in Group I (control), Group II (enoxaparin), Group III (nadroparin) and Group IV (fondaparinux).

**Figure 4 medicina-58-01590-f004:**
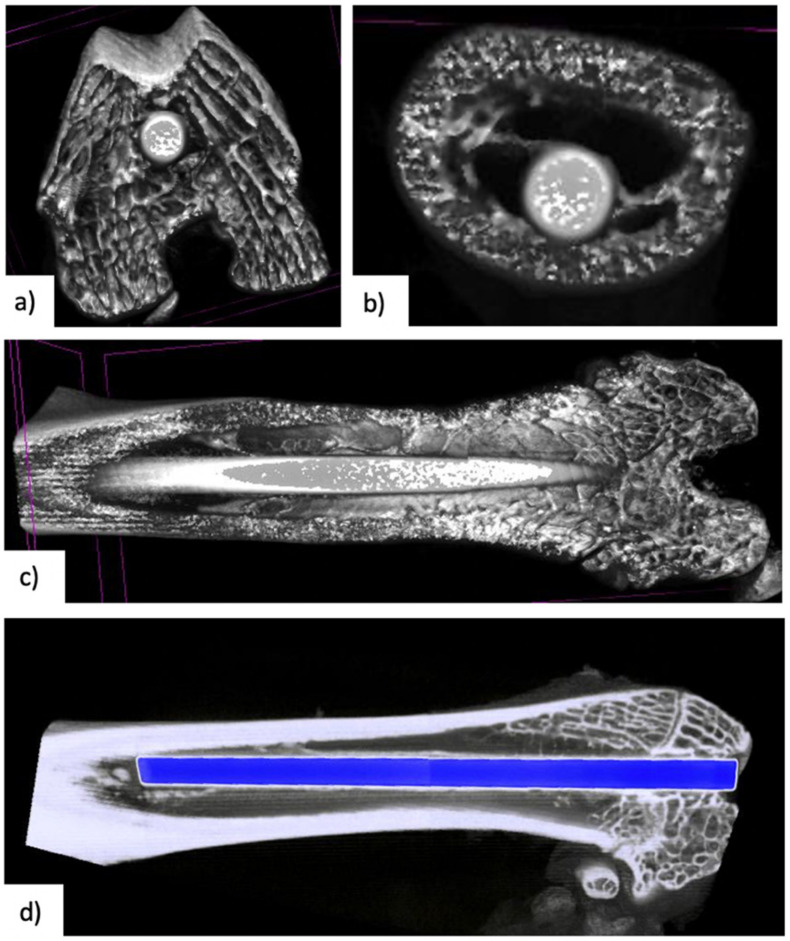
Images obtained following micro-CT examinations: (**a**) cross-sectional view of the intramedullary nail at the distal epiphysis; (**b**) cross-sectional view of the intramedullary nail at the middle of the femoral diaphysis; (**c**) longitudinal view of the intramedullary nail; (**d**) longitudinal view of the intramedullary nail with automatic detection of the metal implant by the software.

**Figure 5 medicina-58-01590-f005:**
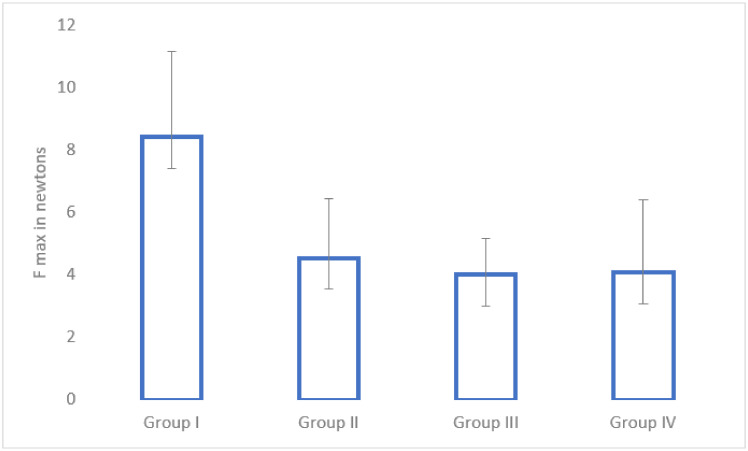
Mechanical pull-put test in Group I (control), Group II (enoxaparin), Group III (nadroparin) and Group IV (fondaparinux).

**Figure 6 medicina-58-01590-f006:**
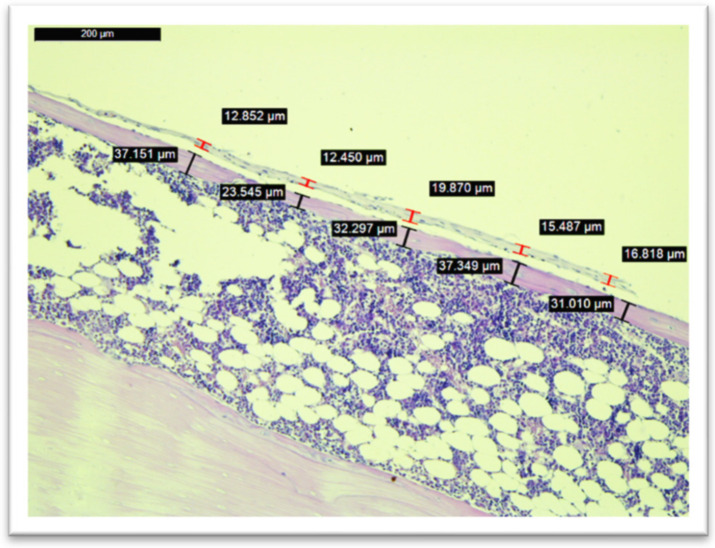
Histological measurements of cortical bone tissue and fibrous tissue in the peri-implant site within Group I (control).

**Figure 7 medicina-58-01590-f007:**
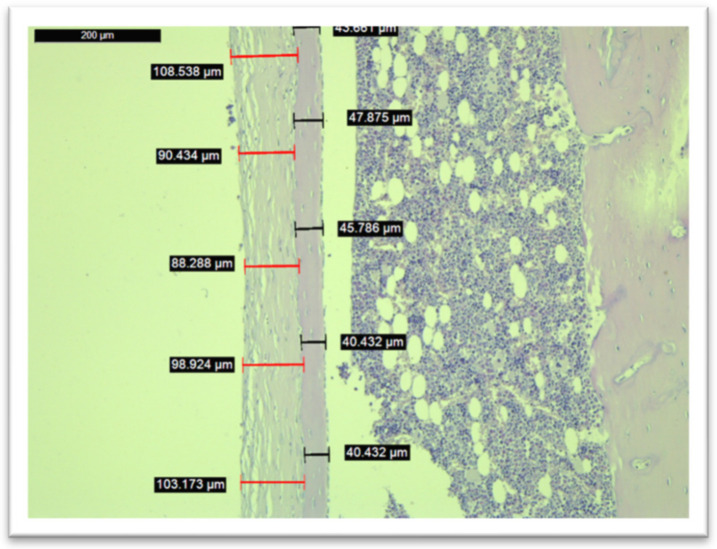
Histological measurements of cortical bone tissue and fibrous tissue in the peri-implant site within Group III (nadroparin).

**Figure 8 medicina-58-01590-f008:**
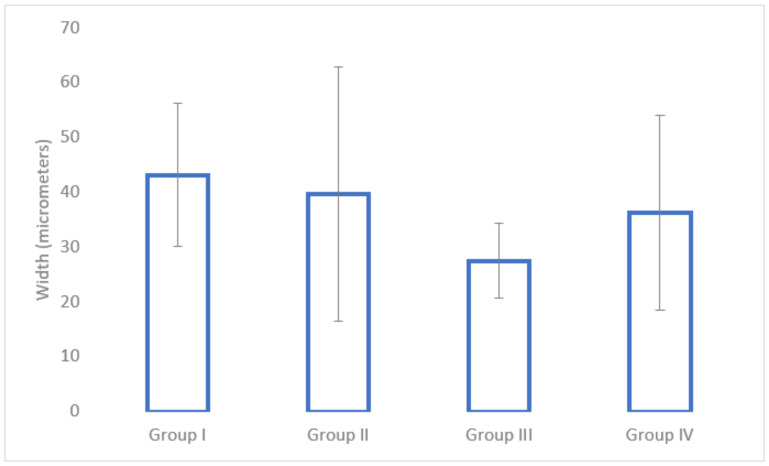
The average width of the cortical bone tissue at the peri-implant region.

**Figure 9 medicina-58-01590-f009:**
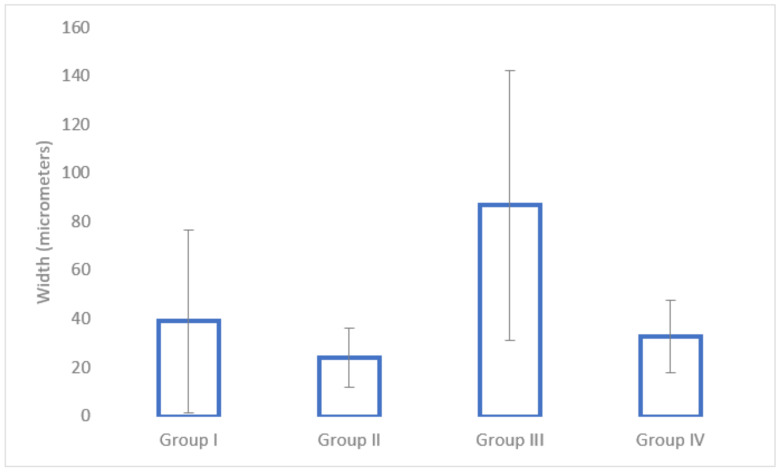
The average width of the fibrous tissue at the peri-implant region.

**Figure 10 medicina-58-01590-f010:**
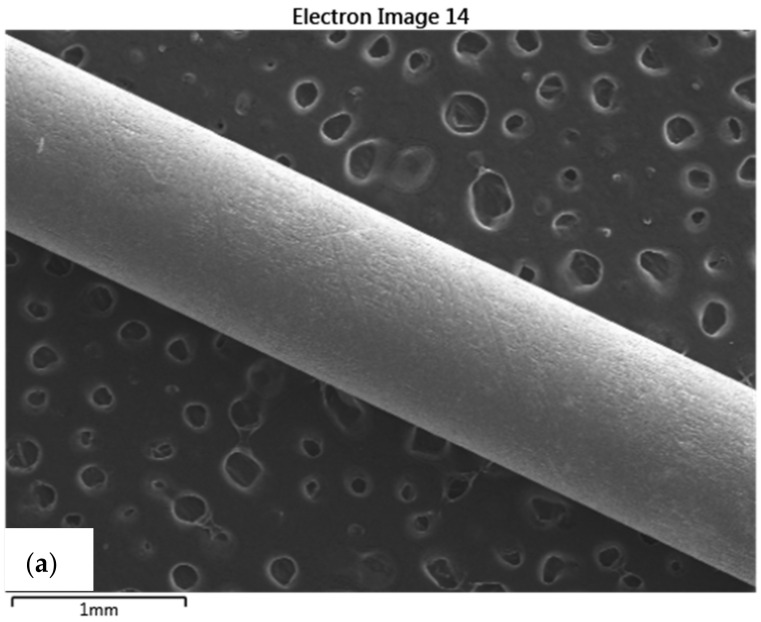
(**a**,**b**) Scanning electron microscopy image at different image magnification; (**c**) EDX analysis diagram.

**Figure 11 medicina-58-01590-f011:**
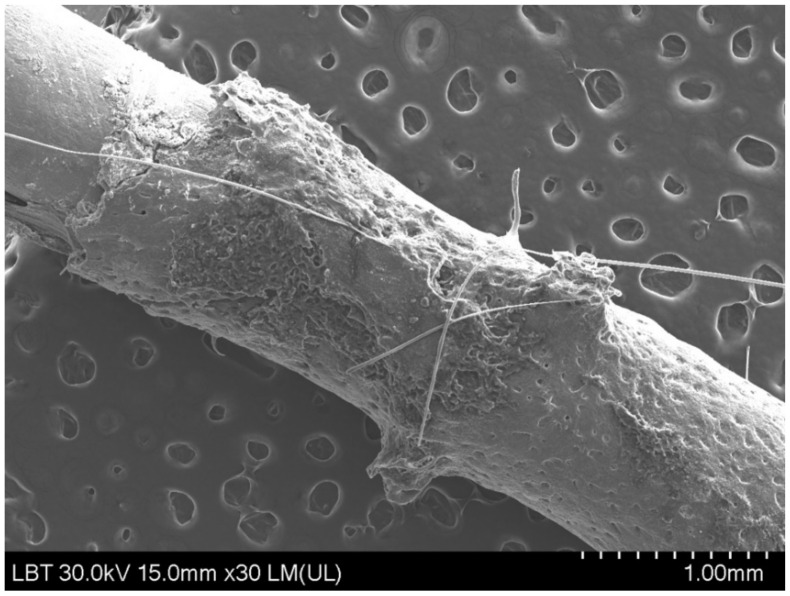
Scanning electron microscopy image of one case with an increased concentration on both calcium and phosphorus.

**Table 1 medicina-58-01590-t001:** Results of micro-CT examination expressed in mean (± standard deviation).

Parameter	Group I (Control) *n* = 3	Group II (Enoxaparin) *n* = 3	Group III (Nadroparin) *n* = 3	Group IV (Fondaparinux) *n* = 3
Bone volume (BV)	0.21 (±0.02) ^b^	0.20 (±0.04)	0.12 (±0.06) ^a^	0.22 (±0.03)
Percent bone volume (BV%)	1.93 (±0.15) ^b^	1.72 (±0.5)	1.46 (±0.2) ^a^	1.75 (±0.4)
Bone surface (BS)	39.46 (±6.5)	38.54 (±7.5)	32.23 (±9.2)	40.21 (±4.8)
Tissue surface (TS)	43.21 (±2.4)	45.32 (±4.2)	56.56 (±7.1)	43.32 (±5.3)
Bone surface/volume ratio (BS/VR)	207.68	226.7	230.21	236.52
Mean total cross-sectional bone area	0.038 (±0.02)	0.034 (±0.025)	0.029 (±0.05)	0.036 (±0.045)
Cross-sectional thickness	0.013 (±0.002)	0.014 (±0.004)	0.010 (±0.004)	0.016 (±0.002)
Trabecular diameter	0.020 (±0.003)	0.023 (±0.0025)	0.016 (±0.005)	0.021 (±0.002)
Trabecular number	8.82 (±0.53)	9.2 (±0.63)	7.52 (±0.78)	8.26 (±0.32)

Note: ^a^ Statistically significant compared to Group I; ^b^ Statistically significant compared to Group IV.

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
