# Peer review of "Low-Molecular-Weight Heparins (LMWH) and Synthetic Factor X Inhibitors Can Impair the Osseointegration Process of a Titanium Implant in an Interventional Animal Study"

_medicina, 2022, doi:10.3390/medicina58111590_

Round 1
Reviewer 1 Report
Dear Authors the paper ”Low molecular weight heparins (LMWH) and synthetic factor X inhibitors can impair the osseointegration process of titanium implant in animal model ‘’is really interesting, well conducted and fits the objectives of the journal; but I suggest to improve review some points
-First, i ask you to check the plagiarism of your article using specific sites to get a similitary report
-About the Title of the article,I suggest you to modify it and add the type of article.
- The introduction section is very short and is needed to add other references to increase the quality of the manuscript, Add recent references about the topic of the article, dwelling in the introduction on articles published in 2022 and describing what your article will add compared to the last articles published; Preferably a published articles should be with 90 or more references.
I suggest you some articles (prothesis, prothesis load, prothesis interferences, and stem cells use for osseointegrations) that will help you improve your article.
Telescopic overdenture on natural teeth: Prosthetic rehabilitation on OFD syndromic patient and a review on available literature PubMed ID 29460531
Prosthodontic Treatment in Patients with Temporomandibular Disorders and Orofacial Pain and/or Bruxism: A Review of the Literature https://doi.org/ 10.3390/prosthesis4020025
Stem Cells in Temporomandibular Joint Engineering: State of Art and Future Persectives. The Journal of Craniofacial Surgery: October 2022 - Volume 33 - Issue 7 - p 2181-2187
doi: 10.1097/SCS.0000000000008771
-I suggest you add a table with the list of abbreviations used in the text.
Thank You,
Kind Regards
Author Response
Dear Reviewer,
First of all, the authors want to thank you for your thorough reading of our manuscript and for your valuable comments. We consider that your comments will increase the overall quality of the manuscript. The comments have been addressed below:
- First, i ask you to check the plagiarism of your article using specific sites to get a similitary report
Response: We have attached the report document generated by the plagiarism software available at our university. The report shows that all the coefficients are low.
- About the Title of the article,I suggest you to modify it and add the type of article.
Response: We have changed the title of the article and added the type of article
- Add recent references about the topic of the article, dwelling in the introduction on articles published in 2022 and describing what your article will add compared to the last articles published; Preferably a published articles should be with 90 or more references. I suggest you some articles (prothesis, prothesis load, prothesis interferences, and stem cells use for osseointegrations) that will help you improve your article. Telescopic overdenture on natural teeth: Prosthetic rehabilitation on OFD syndromic patient and a review on available literature PubMed ID 29460531
Prosthodontic Treatment in Patients with Temporomandibular Disorders and Orofacial Pain and/or Bruxism: A Review of the Literature https://doi.org/ 10.3390/prosthesis4020025 Stem Cells in Temporomandibular Joint Engineering: State of Art and Future Persectives. The Journal of Craniofacial Surgery: October 2022 - Volume 33 - Issue 7 - p 2181-2187 doi: 10.1097/SCS.0000000000008771
Response: We have added the required information in the Introduction section of the manuscript, including the three recommended citations. Also, we have doubled the Introduction section adding more references.
- I suggest you add a table with the list of abbreviations used in the text.
Response: We have added a table with the list of abbreviations found in the text. It can be found ahead of the References section.
Reviewer 2 Report
1. The abstract should be broadened to give additional quantitative results.
2. Please end your abstract with a "take-home" message.
3. Rearrange the keywords so that they are in alphabetical order.
4. What is the current study's novel? It has been extensively researched in the past. Nothing truly novel in its current state. The absence of anything original makes the current study seem like a replication or a modified study. The introduction section should contain specifics about the writers' uniqueness.
5. In order to demonstrate the research gaps that the current study aims to address, previous studies linked to it need to be explained in the introduction part, including their work, their novelty, and their limitations.
6. First paragraph of introduction section needs to be split into two paragraphs, with the second paragraph starting from explaining about aseptic loosening in line 67.
7. The second paragraph after spitted needs more elaboration.
8. In line 68-69, the author mention titanium surfaces? It is jumping. Previous information that related to the present sentence should be added. Please explain it one by one more carefully, not make the explanation become jumping.
9. Toxicity effect of metals materials in titanium alloy needs more explanation. Additionally, the MDPI's suggested reverence should be taken to substantiate this explanation as follows: Jamari, J.; Ammarullah, M. I.; Santoso, G.; Sugiharto, S.; Supriyono, T.; Heide, E. van der. In Silico Contact Pressure of Metal-on-Metal Total Hip Implant with Different Materials Subjected to Gait Loading. Metals (Basel). 2022, 12, 1241. https://doi.org/10.3390/met12081241
10. In the second paragraph of present form, the authors explain about in vivo study. It is needing more explanation regarding in vitro, in vivo, and in silico as general. Also, pointing out the reasons for present study performs in vivo. The suggested reference in comments number 10 would be adopted for supporting this explanation since the paper explain it.
11. Rather than relying just on the predominate text as it already exists, the authors could incorporate more illustrations as figures in the materials and methods section that illustrate the workflow of the current study.
12. What is the basis for patient selection? Is there any protocol, standard, or basis that has been followed? It is unclear since the patient is very heterogeneous with a small number. The resonance involved impacts the present result makes this study flaws. One major reason for rejecting this paper.
13. It's also essential to include additional information on the manufacturer, country, and specifications of the tools.
14. Important information that must be mentioned in the publication relates to the error and tolerance of the experimental equipment utilized in this investigation. As a result of the disparate findings in subsequent research by other researchers, it would be a useful discussion.
15. Outcomes must be compared to similar past research.
16. What is the limitation of the present work? Please include it before the conclusion section.
17. In the conclusion section, further research must be discussed.
18. Five years back literature should be enriched into the reference.
29. In the whole of the manuscript, the authors sometimes made a paragraph only consisting of one or two sentences that made the explanation not clearly understood. The authors need to extend their explanation to become a more comprehensive paragraph. In one paragraph, it is recommended to consist of at least 3 sentences with 1 sentence as the main sentence and the other sentences as supporting sentences. For example in conclusion section.
20. Because of grammatical faults and linguistic style, the authors must proofread the document.
Author Response
Dear Reviewer,
First of all, the authors want to thank you for your thorough reading of our manuscript and for your valuable comments. We consider that your comments will increase the overall quality of the manuscript. The comments have been addressed below:
- The abstract should be broadened to give additional quantitative results.
Response: The Abstract has been modified and includes the most relevant quantitative results.
- Please end your abstract with a "take-home" message.
Response: We have added an additional conclusion to the Abstract section, as a “take-home”message.
- Rearrange the keywords so that they are in alphabetical order.
Response: We have rearranged the keywords in the alphabetical order.
- What is the current study's novel? It has been extensively researched in the past. Nothing truly novel in its current state. The absence of anything original makes the current study seem like a replication or a modified study. The introduction section should contain specifics about the writers' uniqueness.
Response: We have added information about the novely of our study in the Introduction section.
This study aims to test for the first time in the literature the impact of thromboprophylaxis agents on the process of osseointegration in vivo. For this study, we tested three of the most commonly used drugs in DVT prophylaxis: enoxaparin, nadroparin, and fondaparinux, in terms of early implant fixation in vivo.
Our study is completely new, as the effect of LMWHs on titanium implant osseointegration has never been studied before in vivo. We do not even have other studies for us to compare with. Other studies made research of fracture healing and bone metabolism, but not on titanium implant osseointegration. The importance is very high, because these drugs are used in every patient following total hip replacement.
- In order to demonstrate the research gaps that the current study aims to address, previous studies linked to it need to be explained in the introduction part, including their work, their novelty, and their limitations.
Response: We have added new information in the Introduction section to demonstrate the research gaps.
Previous studies by Kock et al. and Osip et al. showed that LMWHs could inhibit osteoblastogenesis during in vitro experimental studies [11, 12, 13, 14]. These studies were the first to prove that low molecular weight heparins have an impact on bone metabolism. One explanation is that LMWHs can alter the function of the cytokines involved in osteoblastogenesis and osteoclastogenesis [15]. As a result, the whole osseointegration process can be affected [15]. Numerous in vivo studies were performed to study the effects of LMWHs on bone biology. Enoxaparin, dalteparin, nadroparin and tinzaparin were shown to increase osteoclastogenesis and bone resorption by modulating M-CSF and TGF- β1, [16-22]. Also, researchers studied the effects of LMWHs on bone metabolism in the case of fracture healing. A study performed by Strett et al. showed that enoxaparin attenuated the bone repair process compared to the control group [x23]. The result is confirmed by a more recent study by Li et al., which concluded that enoxaparin suppresses osteoblastogenesis [24]. On the other hand, other studies showed that LMWHs did not impair the fracture process [25,26].
Although previous studies showed that LMWHs have an impact on bone biology, the overall effect is still controversial. Moreover, we did not find any study to test the effects of LMWHs on the process of osseointegration of the titanium implant. We consider that in vivo studies are essential to test the osseointegration process of titanium implants because there is an implication of osteoclasts, osteoblasts and titanium surface, which are impossible to replicate in the case of in vitro studies. We also consider it essential to know whether thromboprophylaxis agents can impair the titanium implant osseointegration. Clinical trials are challenging to be performed due to a lack of specific examinations for the osseointegration process in the clinical setting, which are available in the animal model. Moreover, the long follow-up to study the rate of aseptic loosening makes clinical trials more difficult to perform.
- First paragraph of introduction section needs to be split into two paragraphs, with the second paragraph starting from explaining about aseptic loosening in line 67.
Response: We have divided the first paragraph into two paragraphs, as recommended.
- The second paragraph after spitted needs more elaboration.
Response: We have elaborated the second paragraph to provide more information about aseptic loosening.
Aseptic loosening can be prevented with a more enhanced osseointegration process, represented by bone apposition at the titanium surface of the total hip arthroplasty implant [5,6]. This complex process is dependent on the processes of bone formation, performed by osteoblasts, and bone resorption, performed by osteoclasts. The process of ossointegration is regulated by many cellular pathways which modulate the activity of osteoblasts and osteoclasts [6]. Osteoblasts arise from mesenchymal stem cells (MSC) under the influence of cytokines, such as tumor necrosis factor (TNF) alpha, interleukin (IL) 1, IL-6 and IL-11. On the other hand, osteoclastogenesis is positively modulated by M-CSF and TGF-β1. The more active the osteoblasts are compared to osteoclasts, the stronger the implant fixation will be and the lower risk of aseptic loosening. The process of osseointegration is similar in the case of titanium intramedullary implants and titanium dental implants [7, 8, 9]. Our study group has proven that osseointegration can be influenced by many factors, including systemic drugs [10].
- In line 68-69, the author mention titanium surfaces? It is jumping. Previous information that related to the present sentence should be added. Please explain it one by one more carefully, not make the explanation become jumping.
Response: We have added more information about the titanium surfaces to make the introduction section more fluent.
Total hip replacement is a common procedure performed worldwide to treat hip osteoarthritis. This pathology affects 10 to 13% of people over 60 years old [1]. Total hip replacement implants are made from titanium alloys. The most frequently-used titanium alloy in total hip replacements is Ti90Al6V4, consisting of titanium, aluminium and vanadium. Although offering good results overall, complications of total hip replacement exist. The most frequent late complication of this type of this surgical procedure is a deficient implant fixation, called aseptic loosening, which leads to increased pain and disability [2]. Patients affected by this complication cannot weight-bear on the affected limb, thus leading to an important functional deficit. When aseptic loosening is present, revision surgery is required, which is expensive for the healthcare system [3]. Moreover, it is technically demanding, requires an experienced team, and is usually performed in tertiary care hospitals. Also, the revision of the total hip replacement often results in a lower patient satisfaction rate than the primary hip replacement [4].
Aseptic loosening can be prevented with a more enhanced osseointegration process, represented by bone apposition at the titanium surface of the total hip arthroplasty implant [5,6]. This complex process is dependent on the processes of bone formation, performed by osteoblasts, and bone resorption, performed by osteoclasts.
- Toxicity effect of metals materials in titanium alloy needs more explanation. Additionally, the MDPI's suggested reverence should be taken to substantiate this explanation as follows: Jamari, J.; Ammarullah, M. I.; Santoso, G.; Sugiharto, S.; Supriyono, T.; Heide, E. van der. In Silico Contact Pressure of Metal-on-Metal Total Hip Implant with Different Materials Subjected to Gait Loading. Metals (Basel). 2022, 12, 1241. https://doi.org/10.3390/met12081241
Response: The article cannot be added because it uses meta-on-metal hip implants, which are very less used today worldwide due to their high number of complications. For example, I send the FDA website where they state “To date, there are no FDA-approved metal-on-metal total hip replacement devices marketed for use in the US.” https://www.fda.gov/medical-devices/implants-and-prosthetics/metal-metal-hip-implants
Also, the toxicity of metal implants has no impact on the osseointegration process.
- In the second paragraph of present form, the authors explain about in vivo study. It is needing more explanation regarding in vitro, in vivo, and in silico as general. Also, pointing out the reasons for present study performs in vivo. The suggested reference in comments number 10 would be adopted for supporting this explanation since the paper explain it.
Response: We have added more information in the Introduction section about the in vivo and in vitro studies, as well as why we opted for in vivo study.
Although previous studies showed that LMWHs have an impact on bone biology, the overall effect is still controversial. Moreover, we did not find any study to test the effects of LMWHs on the process of osseointegration of the titanium implant. We consider that in vivo studies are essential to test the osseointegration process of titanium implants because there is an implication of osteoclasts, osteoblasts and titanium surface, which are impossible to replicate in the case of in vitro studies. We also consider it essential to know whether thromboprophylaxis agents can impair the titanium implant osseointegration. Clinical trials are challenging to be performed due to a lack of specific examinations for the osseointegration process in the clinical setting, which are available in the animal model. Moreover, the long follow-up to study the rate of aseptic loosening makes clinical trials more difficult to perform.
This study aims to test for the first time in the literature the impact of thromboprophylaxis agents on the process of osseointegration in vivo. For this study, we tested three of the most commonly used drugs in DVT prophylaxis: enoxaparin, nadroparin, and fondaparinux, in terms of early implant fixation in vivo.
- Rather than relying just on the predominate text as it already exists, the authors could incorporate more illustrations as figures in the materials and methods section that illustrate the workflow of the current study.
Response: We have added Figure 2 in order to better explain the timeline of our project.
- What is the basis for patient selection? Is there any protocol, standard, or basis that has been followed? It is unclear since the patient is very heterogeneous with a small number. The resonance involved impacts the present result makes this study flaws. One major reason for rejecting this paper.
Response: The animals used within the study and their selection was described in the Animal model section of Materials and Methods:
The study received approval from the Ethical Commission of the local university (no. 210/02/04/2020). The experiments were performed at the Center of Experimental Medicine Cluj-Napoca and according to the European guidelines (directive 2010/63/EU). 70 female albino Wistar rats of 8-10 weeks old and with a weight of 190 ± 30 mg were used. The animals were raised at the same animal facility without any genetic modification, while food and water were provided ad libitum. A veterinary doctor checked all of the subjects to be enrolled in the study to be clinically healthy. The subjects were randomized into four groups: Group I (OVX group, n=22), Group II (OVX + enoxaparin, n=16), Group III (OVX + nadroparin, n=16), and Group IV (OVX + fondaparinux, n=16).
We have performed many animal studies so far, and to our knowledge, there is no other way to make the animal subject selection. All of the rats are from the same breed, similar age and weight and no genetic modifications. Moreover, all of the studies performed at the animal facility use the same protocol as described before.
- It's also essential to include additional information on the manufacturer, country, and specifications of the tools.
Response: We have added information about the manufacturer, country and relevant specfifications of the tools.
- Important information that must be mentioned in the publication relates to the error and tolerance of the experimental equipment utilized in this investigation. As a result of the disparate findings in subsequent research by other researchers, it would be a useful discussion.
Response: There is no information to be found in research articles regarding the error or tolerance of the experimental equipment used in the investigation. We used only equipment specially designed for research purposes, in order to provide the lowest error.
We have added the following text in the Discussion section regarding the comment:
Bruker Skyscan 1172 micro-CT is a microfocus X-ray microtomography optimized for small samples offering a good precision for osseointegration examination. The error and tolerance are insignificant since a standard method of determining the region of interest was used in all samples. The Zwick/Roell Z005Ò testing machine has a machine compliance correction, offering real-time modifications for the highest possible level of precision.
- Outcomes must be compared to similar past research.
Response: There are no similar past researches for our study. It is the first time any study tests in vivo the osseointegration process of titanium implants under the influence of thromboprophylaxis agents. The previous studies were in vitro and tested the influence on the osteoblast and osteoclast cultures. Our study has other determinations (micro-CT, histology, pull-out test, serum analysis). We have included a comparison to fracture healing in the Discussion section, as it is the closest study to our current manuscript.
We have found no similar studies to compare our study’s results. Nevertheless, other studies on bone metabolism showed that enoxaparin and nadroparin increase osteoclastogenesis, thus leading to an increased bone resorption process [16-22]. These results are according to our study, where the osseointegration process is impaired by enoxaparin and nadroparin. Other studies on fracture healing have shown that enoxaparin can impair bone growth compared to control, a result related to our study due to a deficient bone metabolism [23].
- What is the limitation of the present work? Please include it before the conclusion section.
Response: The study has the following limitations, which have been included in the discussion section:
The study also has some limitations. The main limitation of our study is the inability to perform the histological analysis with the implant in situ. This could provide a better assessment of the histological bone-implant contact. During the implant removal, even though it is performed at low speeds, a quantity of bone and fibrous tissue could still be attached to the implant and therefore provide deficient information when the bone specimens are analyzed. In order to test this hypothesis, we performed an SEM/EDX analysis which showed that only a small quantity of calcium and phosphorus had been removed along with the implants, with only one exception, which was later excluded. This result provides sufficient information to state that the implant removal during the pull-out test at low speed does not affect the histological analysis of the implant site.
Another limitation is the lack of a known therapeutic range for the anti-Xa coagulation factor level for us to know whether or not the dosing was correct. This limitation could influence the study’s results due to potentially different concentrations of drugs. Also, a limitation of our study is the relatively low number of subjects for micro-CT analysis and SEM/EDX analysis.
- In the conclusion section, further research must be discussed.
Response: We have added information about further research in the Conclusion section.
Short-term administration of enoxaparin, nadroparin, and fondaparinux can reduce the osseointegration of titanium implants, while nadroparin resulted in the highest quantity of fibrous tissue and the lowest quantity of cortical bone tissue surrounding the implant site. Further clinical research is needed to test the influence of thromboprophylaxis agents on the process of osseointegration.
- Five years back literature should be enriched into the reference.
Response: We have only found one study in the last five years which has importance to our study, and we have included them in the study.
Li, Y.; Liu, L.; Li, S.; Sun, H.; Zhang, Y.; Duan, Z.; & Wang, D. (2022). Impaired bone healing by enoxaparin via inhibiting the differentiation of bone marrow mesenchymal stem cells towards osteoblasts. Journal of bone and mineral metabolism, 40(1), 9–19.
- In the whole of the manuscript, the authors sometimes made a paragraph only consisting of one or two sentences that made the explanation not clearly understood. The authors need to extend their explanation to become a more comprehensive paragraph. In one paragraph, it is recommended to consist of at least 3 sentences with 1 sentence as the main sentence and the other sentences as supporting sentences, for example, in the conclusion section.
Response: We have reviewed the entire manuscript and made changes throughout the text to increase clarity and readability.
- Because of grammatical faults and linguistic style, the authors must proofread the document.
Response: We have had the document proofread and made the changes throughout the manuscript.
Round 2
Reviewer 2 Report
Reviewers greatly appreciate the efforts that have been made by the author to improve the quality of their articles after peer review. I reread the author's manuscript and further reviewed the changes made along with the responses from previous reviewers' comments. Unfortunately, the authors failed to make some of the substantial improvements they should have made making this article not of decent quality with biased, not cutting-edge updates on the research topic outlined. In addition, the author also failed to address the previous reviewer's comments, especially on comments number 4 (noting something really novel), 5 (noting ground-breaking explanation), 9 (suggested literature not incorporated), and 11 (not explained well with really basis). With all due respect, the reviewer opposed this article to be published and must be rejected. Thank you very much for the opportunity to read the author's current work.